# Bioremediation Potential of Native *Bacillus* sp. Strains as a Sustainable Strategy for Cadmium Accumulation of *Theobroma cacao* in Amazonas Region

**DOI:** 10.3390/microorganisms10112108

**Published:** 2022-10-25

**Authors:** Marielita Arce-Inga, Alex Ricardo González-Pérez, Elgar Hernandez-Diaz, Beimer Chuquibala-Checan, Antony Chavez-Jalk, Kelvin James Llanos-Gomez, Santos Triunfo Leiva-Espinoza, Segundo Manuel Oliva-Cruz, Liz Marjory Cumpa-Velasquez

**Affiliations:** 1Laboratorio de Sanidad Vegetal, Instituto de Investigación para el Desarrollo Sustentable de Ceja de Selva (INDES-CES), National University Toribio Rodriguez de Mendoza de Amazonas (UNTRM), Calle Higos Urco 342, Chachapoyas 01001, Amazonas, Peru; 2Laboratorio de Microbiología Ambiental y Extremófilos, Departamento de Ciencias Biológicas y Biodiversidad, Universidad de los Lagos, Avda Fuchsclocher 1305, Osorno 5290000, Chile

**Keywords:** bioaccumulation, cadmium-hypertolerant bacteria, Amazonian cacao

## Abstract

Cacao plant cadmium accumulation has become a major concern, especially for small Amazonian producers. A sustainable alternative to address its toxicity is the use of cadmium removal bacteria. In this regard, 138 rhizosphere isolates from cacao were examined. Supported by their phenotypic characterization and in vitro cadmium tolerance, three hypertolerant bacteria were selected and identified as members of the *Bacillus* (S1C2, R1C2) and *Pseudomonas* (V3C3) genera. They were able to grow normally and reduce the cadmium content under in vitro conditions. However, only S1C2 and R1C2 evidenced to employ intracellular Cd^2+^ accumulation, suggesting the variability of bacterial detoxification mechanisms. Their bioremediation capacity for *Theobroma cacao* CCN51 was also analyzed. Surprisingly, we found high detectable levels of Cd^2+^ in the non-cadmium supplemented control, suggesting an extra source of cadmium in the pot. Moreover, despite their cadmium reduction performance under in vitro conditions, they exerted highly variable outcomes on stem cadmium accumulation. While S1C2 and R1C2 showed a considerable reduction of Cd content in cacao stems, the strain V3C3 did not show any effect on Cd content. This highlights the complexity of the plant–bacteria interactions and the importance of the in vivo test for the selection of promising PGPR bacteria. Overall, our results suggest the cadmium alleviation potential and promising prospects of native *Bacillus* strains associated with Amazonian cacao.

## 1. Introduction

Cadmium (Cd^2+^) is a heavy metal present in the environment, derived from parent materials and anthropogenic activities such as mining, sewage, biosolid, fertilizer, or industrial sources [1,2]. It belongs to the heavy transition metals group (Group IIB of the periodic table) and several reports have shown its harmful effects on the human body. Cadmium exposure is mainly through inhalation and ingestion, but the symptoms depend on the dose, exposure time, diet, age, and gender [1,3]. It can be absorbed by multiple human organs leaving effects such as hepatotoxicity, renal dysfunction, skeletal lesions (Itai-itai disease), and lung damage [1,4]. Furthermore, it has shown carcinogenic potential and epigenetic modifications in DNA [2,5]. Therefore, several actions have been adopted by regulatory agencies to protect the consumer from the consumption of cadmium-contaminated products [6,7,8].

Although cereals and vegetables are the major sources of cadmium [3,9], the highest concentrations can be also found in certain luxury foods such as chocolate. This is produced from cocoa beans of *Theobroma cacao* L., a perennial tree native to the Amazonian rainforest of South America and cultivated in the world’s tropics. In the world cocoa market, there are four major groups of the crop: Forastero, Trinitario, Criollo, and National, whose cocoa beans can be categorized in two broader types: the “bulk” or “ordinary” and the “fine aroma” cocoa beans. The latter is highly appreciated for its organoleptic properties provided by a complex flavour profile that reflects the genetics of the crop, the expertise of the producer, and the particular environment where the cacao has been grown, fermented, and dried [10].

Despite the beneficial effects associated with moderate chocolate consumption [11,12], it has been seen that some cocoa-based food products, including chocolate, exceed the EU and Chinese maximum accepted Cd^2+^ contaminant level thresholds [13]. Although the Cd^2+^ contents seem to be variable, high levels of contamination can be found in the cacao beans, powder, or dark chocolate, with an increasing trend, depending on the solid cocoa content [14].

Cadmium is introduced into the food chain through plant uptake. According to [15], the cadmium concentrations in crops depend on their availability in the soil. This statement has also recently been supported in cacao. A meta-analysis of over 400 studies on soil–plant Cd^2+^ relationships has already shown that cacao bean Cd^2+^ increases almost proportionally to the total soil Cd^2+^ content [16]. Moreover, the geographical origin is also an important factor that explains Cd^2+^ contamination in cacao beans, as demonstrated by a multi-element traceability test among 23 countries which found elevated levels of Cd^2+^ in cacao bean samples, especially those coming from South America [17]. Similar findings were reported by [18,19,20,21].

Cacao is an economically and socially important crop in Perú, especially in the Amazonas region. According to [22], in recent years the national production of cocoa beans has been increasing steadily. Furthermore, it has replaced illegal crops, becoming the major support of at least 90,000 Amazonian families. However, with the set limit implementation by regulatory agencies, the cadmium content level has become a concerning situation to the Amazonas organic producers. Samples analysis from the major Peruvian cacao producer regions overpasses the maximum cadmium values of the international standard limits set (0.80 μg/g) in the Commission Regulation (EU) N° 488/2014 [6]. For example, cocoa bean samples from Piura, Tumbes and, Huánuco have shown Cd^2+^ values over 0.96 μg/g [23,24]. More recently, [25] reported elevated cadmium contents in root, leaf, testa, and cotyledon samples in 59% of the plots were evaluated in the province of Bagua, department of Amazonas.

A sustainable alternative to address cadmium soil toxicity is the use of beneficial microorganisms. Plant growth-promoting rhizobacteria (PGPR) can be regarded as an efficient, inexpensive, and more environmentally friendly tool to improve growth or to solve some specific agronomic issues [26]. For heavy metal alleviation, an increasing number of studies have reported the isolation and identification of PGPR with heavy metal removal abilities [27]. Particularly, in cadmium-contaminated soils, cadmium-tolerant bacteria (CdtB) have been shown to play a pivotal role [28,29]. They have the capacity to interact with Cd in the soil, even at toxic levels, and use it as a source of energy to activate various mechanisms [29,30]. Among them, the major mechanisms are biosorption and intracellular accumulation; these involve passive cell surface metal binding and active heavy metal uptake, respectively [31,32]. The genetic determinants can be found either on plasmids or chromosomes and may vary according to bacterial species. Recently, the genetic background of the processes involved has been nicely reviewed by [33].

Several CdtB, in association with different vegetable models, have been described as efficient cadmium removal agents [27,29,34] In cacao, the CdtB Enterobacter CdDB41 isolated from cacao soils showed a major Cd immobilization rate under in vitro conditions [28]. Similar results have been seen for the strain *Klebsiella variicola* 18-4B [35]. Moreover, recently, the application of the preformulated inoculum of the strain *Bacillus* CdTB14 showed a decrease of 0.30 + 0.1 mg kg^−1^ of soil Cd [36]. Therefore, the use of heavy metal-resistant bacteria has received more attention in recent years. However, the use of PGPR is not always successful for several reasons, including lack of nutrients in the soil or competition with indigenous communities [37]. Because of this, the utilization of native PGPR adapted to the local environment with the appropriate metabolic properties is a more promising approach [38]. The aim of this study was to identify native rhizobacteria with cadmium bioremediation potential associated with the *T. cacao* cultivated in cadmium-contaminated soils in the Amazonas department of Perú.

## 2. Methodology

### 2.1. Soil Samples Collection

The soil samples were taken from the rhizosphere, the soil layer adjacent to the root, of *T. cacao* accessions at 20 cm deep from different plots in three districts of Bagua province: Copallin (5°36′30.38″ S–78°21′08.40″ W), Aramango (5°24′50.77″ S–78°25′47.25″ W) and La Peca (5°35′05.67″ S–78°26′27.05″ W), whose cadmium soil levels were above 1.4 ppm (Appendix A). This was based on the maximum cadmium content allowed in agriculture soils by the Environment’s Ministry of Perú [39]. Approximately 200 g of rhizosphere soil, from each plant, was carried out in plastic sterile bags to the laboratory and stored at 4 °C until processing.

### 2.2. Bacterial Isolation

Bacterial strains were obtained from each rhizosphere soil sample. For this, 20 g of soil was homogenized by agitation in 180 mL of sodium chloride solution 0.85% (*w*/*w*). Serial dilutions were plated in Luria-Bertani (LB) agar medium supplemented with 3 ppm of CdCl_2_ (micro toxicity condition) and incubated at 28 °C for 24 h until the formation of visible single colonies. For preservation, a single colony was grown in LB liquid medium at 28 °C for 24–48 h, mixed with glycerol 30% (*v/v*), and stored at −80 °C.

### 2.3. Cadmium Tolerance Phenotypic Assay and Morphological Characterization

The cadmium tolerance was evaluated in LB medium supplemented with 50, 100, 150, 200, 300, 350, and 400 ppm of CdCl_2_ throughout the minimum inhibitory concentrations method (MIC). An aliquot of 10 μL, previously grown until exponential phase 10^7^ colony forming units (CFU/mL) in LB broth, was streaked in LB medium at different Cd^2+^ concentrations and incubated at 28 °C for 24 h. Those with visible growth were considered positive. The morphological characterization was made as indicated by [40]. The variables recovered were shape, color, margin, and elevation. The Gram staining procedure was also performed.

### 2.4. BOX Fingerprinting and 16S rRNA Phylogenetic Identification

Genomic bacterial DNA was extracted and purified using the Wizard^®^ DNA extraction Kit (*Promega*) according to the manufacturer’s instructions. Both BOX-PCR fingerprinting and 16S rRNA gene amplification were carried out using BOX-A1R, 27F, and 1492R primers [41,42]. The BOX-PCR products were analyzed as previously described by [43]. For phylogenetic and molecular evolutionary analyses, the resulting 16S rRNA amplicons were sequenced (Macrogen, Seoul, South Korea) and aligned with reference type sequences obtained from the National Center for Biotechnology Information (NCBI) Genbank Database [44] and The Ribosomal Database Project (RDP) [45] using MEGA software version 11.0 [46]. Distances were calculated using a complete-deletion procedure, phylogenies were built using the maximum-likelihood method, and the robustness of tree topologies was evaluated by bootstrap analysis (1000 replicates). According to the preliminary similarity test, representative type strains sequences belonging to *Pseudomonas* genus and *Bacillaceae* family were included. Only branches supported by a cut-off value of 50% were considered. The 16S rRNA sequences have been deposited in the GenBank database as the following accession numbers: S1C2(ON479211), T3C1(ON479212), R1C2(ON479210), V3C3(ON982449).

### 2.5. In Vitro Removal Cadmium Efficiency

Bacterial strains were grown from a single colony in a 50 mL flask of LB liquid medium supplemented with 100, 200, and 300 ppm of CdCl_2_ without any pH adjustment for 24 h, following similar culture conditions described by [47]. LB medium without Cd^2+^ was used as a control. The supernatant was obtained by centrifugation at 4 °C, 12,000 rpm for 5 min. For cadmium quantification in the medium, 10 mL of supernatant was filtered through a 125 mm filter and measured by atomic absorption spectrophotometry (Agilent Technologies 4100 MP-AES), according to the standard protocols of the Soil Analysis Laboratory of INDES-CES, UNTRM. The removal cadmium efficiency (RE) was calculated as follows: RE% = ((C_0_ − C_f_/C_0_) × 100), where C_0_ is the initial cadmium concentration in the medium and C_f_ is the cadmium concentration in the filtered solution [48].

### 2.6. Intracellular Cadmium Bioaccumulation Capacity

The pellets obtained in the previous step were washed three times with NaCl_2_ 0.85% (*w/w*) and dried at 70 °C. The dried pellets were mixed with 2 mL of HNO_3_/H_2_O_2_ 5:1 (*v/v*) and digested at 70 °C for 6 h. Then, 10 mL of deionized H_2_O was added and centrifuged at 12,000 rpm, 4 °C, for 5 min. The supernatant was filtered, diluted, and used for cadmium determination previously described. The intracellular cadmium accumulation was calculated as CdA = (C_Cd_ × V)/W, where C_Cd_ is the cadmium concentration in the filtered solution, V is the supernatant volume, and W is the dried pellet weight [47].

### 2.7. Bacterial Growth Kinetic

For kinetic growth, bacterial strains were grown in LB liquid medium supplemented with 0, 100, 200, and 300 ppm of CdCl_2_ for 24 h. Then 100 μL aliquots were taken at 2, 4, 6, 8, 10, 12, and 24 h, plated by serial dilution in LB medium, and incubated at 28 °C. After 24 h, the CFU/mL was calculated.

### 2.8. Plant Growth Promoting Traits Characterization

For PGPR traits characterization, bacterial isolates were previously grown in LB medium at 28 °C for 24 h. Then, an aliquot of 10 μL at OD600 = 0.5 was inoculated in triplicate in NBRIP, Chrome Azurol S (CAS), and a Tris-minimal agar-medium as previously described by [43,49]. After 24–48 h of incubation, the variables solubilization index (SI), solubilization efficiency (SE), and siderophore production area were measured.

### 2.9. Growth Plant Promotion Capacity under Cadmium Conditions

The hypertolerant cadmium bacteria S1C2, R1C2, V3C3, and the low-tolerant cadmium strain T3C1 were selected for individual inoculation in cacao seeds. A non-inoculated treatment was used as a control (NB). The bacterial strains were grown in LB liquid medium at 20 °C for 24 h up to the exponential growth phase (approximately 10^7^ CFU/mL). *Theobroma cacao* genotype CCN-51 was used for the current analysis. Seeds, previously germinated in sterile water, were dipped in the bacterial inoculum for 40 min before planting. The pots were prepared with native Amazonian soil with undetectable Cd^2+^ level, previously sterilized at 121 °C for 30 min, three times, and irrigated with 1.4 and 3.5 ppm of CdCl_2_ solution until reaching field capacity. The seedlings were transferred to plastic pots, containing 1000 g of the sterilized soil, in a completely randomized experimental design, and grown, under semi-controlled conditions in the INDES-CES greenhouse installation located in Bagua province (5°44′02.65″ S–78°25′00.91″ W). After 12 weeks, the cacao plants were harvested and carefully dissected into stems and roots. The fresh weight values (FW) of the plant parts were obtained. Then, stems and roots were dried in paper bags at 70 °C until constant weight. Stem and root cadmium concentrations were calculated by atomic absorption spectrophotometry. For this purpose, 1 g of dry material was reduced to ashes at 450 °C for 8 h. Then, the samples were carefully hydrated with distilled water, mixed with 1 mL of HCl, and incubated at room temperature for 1 h, twice. Finally, the samples were filtered and diluted in 25 mL of distilled water.

### 2.10. Data Analysis

Analysis of variance (ANOVA) was performed using Infostat statistical software [50]. One-way ANOVA tests were performed for all the analyses, as indicated in the figure legends. When departure from the homogeneity of residues and/or normality was detected, data transformation procedures (*Log* and Normal Score) were used to improve them. Post hoc comparisons were carried out with Tukey’s multiple-comparison test.

## 3. Results

### 3.1. Cadmium Tolerant Bacteria

A total of 138 bacterial strains, in LB agar medium supplemented with 3 ppm of CdCl_2_ (micro toxicity), were isolated from rhizosphere soil samples collected in three districts of Bagua province, whose cadmium concentrations were above the maximum allowable limit of 1.4 mg/kg set by the Ministry of Environment of Perú. Each isolate was tested to analyze its tolerance under different cadmium toxicity conditions. The test showed that more than 88% (122) of isolates were able to grow normally up to 300 ppm, except for the low-tolerant strain T3C1, which grew until 50 ppm. At 350 ppm, the tolerant population was 36.96% (51) whereas at 400 ppm, only 2.17% (3) were able to develop visible growth (Appendix A). The latter group was classified as hypertolerant-cadmium bacteria (HCdB) and was selected for the removal efficiency and the cadmium accumulation capacity tests. In addition, to explore the morphological diversity, strains that were able to grow over 350 ppm were characterized. Single colonies were morphologically described according to their shape, color, margin, elevation, and Gram stain (Figure 1). We found that 73% of the population was made up of gram-positive bacteria. Moreover, the colonies of these bacteria were predominantly circular (73.1%), convex (51.9%), not pigmented (63.5%), and had entire margins (86.8%) (Table 1).

### 3.2. Bacterial Identification

For comparative purposes the low-tolerant strain T3C1 was chosen, to be included in the plant growing test under cadmium conditions. To avoid genetic redundancy, a fingerprinting test was done by BOX-PCR method. The V3C3, S1C2, R1C2, and T3C1 isolates showed to be genetically different (Appendix A). The 16S rRNA analysis revealed that the strains S1C2, R1C2, and T3C1 belong to the *Bacillaceae* family, specifically to the *Bacillus* and *Peribacillus* genera, respectively (Figure 2 and Appendix A). Our strains are phylogenetically placed along with representative members of the genus *Bacillus* (*Subtilis* and *Cereus* Clade) and the recently proposed genera *Metabacillus*, *Peribacillus*, *Cytobacillus*, *Mesobacillus*, and *Neobacillus* [51]. The strain S1C2 grouped together with *B. wheihenstephanensis* DMS11821^T^ (*Cereus* Clade), the strain R1C2 grouped closely to *B. pumilus* NRRL NRS-272^T^ (*Subtilis* Clade), while the strain T3C1 grouped in the same clade than *Peribacillus simplex* DSM1321^T^, with 62, 100 and 58% bootstrap support respectively (Figure 2A and Appendix A). On the other hand, the strain V3C3 within *P. putida* complex close to *Pseudomonas reidholzensis* CCOS 865^T^ (Figure 2B), with a bootstrap value of 56 % (Figure 2B). Furthermore, the percent similarity was also calculated for each isolate. The S1C2, R1C2, and T3C1 strains were 98.97%, 100%, and 94.72% similar to *B. weihenstephanensis* DMS11821^T^; *B. pumilus* NRRL NRS-272^T^ and *Peribacillus simplex* DSM1321^T^, respectively, while the V3C3 strain was 98.86% similar to *P. reidholzensis* CCOS 865^T^. Despite this, in some cases, stronger bootstrap values were obtained, and in order to resolve up to species identity, the use of complementary phylogenetic markers were needed.

### 3.3. Bacterial Behavior under Cadmium Toxicity and Complementary PGPR Traits

To corroborate the tolerance capacity, the viable bacterial cells (CFU/mL) were quantified at 0, 2, 4, 6, 8, 10, 12, and 24 h. Although some differences among strains were observed, we could find viable cells in all the treatments. The strains V3C3 and S1C2 showed similar growth at 0, 100 and 200 ppm with a slight decrease at 300 ppm CdCl_2_ while the strain R1C2 showed the lowest viable cells at 100, 200, and 300 ppm compared with the control (0 ppm). Despite that, we found over 10^8^ CFU/mL, in all the conditions (Figure 3).

All the strains showed RE values between 29.29 and 37.88% with no statistical differences at 100 ppm. However, *Bacillus* sp. R1C2 and *Pseudomonas* sp. V3C3 were significantly different at 200 and 300 ppm (Figure 4a). The R1C2 strain showed an RE value of 40.38% at 200 ppm while the V3C3 strain showed significant values of 44.03 and 46.04% at 200 and 300 ppm CdCl_2_ compared to S1C2, respectively. This result agrees with the growing behavior described above, where V3C3 showed a better growing capacity than R1C2 and S1C2.

On the other hand, the CdA of the V3C3 strain, as opposed to R1C2 and S1C2, was non-significant with increasing concentrations of Cd^2+^ (Figure 4b). Moreover, R1C2 significantly increases almost 4-fold the cadmium accumulation with the increase in Cd^2+^ initial concentration (Figure 4b). The CdA values were not relevant in the control treatment (0 ppm). These results suggest that V3C3, R1C2, and S1C2 selected strains may efficiently remove cadmium under in vitro conditions.

Complementary PGPR-traits were measured. According to phenotypic tests, the strains R1C2, S1C2, and V3C3 were able to solubilize zinc and phosphate in a similar way. However, only the strain S1C2 was able to produce siderophores under in vitro conditions (Appendix A).

### 3.4. Effect of Selected Strains Inoculation on the Growing Variables and Cadmium Accumulation of Cacao

No significant differences were found in terms of growth (Table 2), except at 3.5 ppm where the post hoc test showed that the strain V3C3 reduced significantly the stem fresh weight (FW_stem_) compared with the control (NB). Despite these results, a positive trend in the bioremediation activity of some strains was found for all variables (FW_stem_, FW_roots_, DW_stem_, and DW_root_) at the control and 1.4 ppm of Cd^2+^. Not-relevant levels were found for root Cd-accumulation. For stem Cd^2+^ content quantification, we surprisingly found high levels of Cd^2+^ in the control treatment (Figure 5). The NB treatment showed 1.8 ± 0.24 ppm of Cd^2+^ while the seedlings inoculated with the low-tolerant strain T3C1 showed 2.82 ± 0.34 ppm, suggesting potential cadmium stimulation to the plant by the strain T3C1. The strain V3C3 was similar to NB. In contrast, strains S1C2 and R1C2 showed a negative trend in stem cadmium accumulation. They showed a considerable cadmium reduction level, but non-statistical significance was found. At 1.4 and 3.5 ppm, no significant differences were found (Figure 5).

## 4. Discussion

Microorganisms play a pivotal role in the main biogeological cycles in the environment. Through different mechanisms such as exclusion, active removal, biosorption, precipitation, or bioaccumulation (both in external and intracellular spaces) they colonize several habitats, even in the presence of some toxic heavy metals. Moreover, some microbes can change the valence states of metals which may convert them into less toxic forms [52]. As a result, in recent years, the use of microorganisms has received more attention.

Microbe-based technologies can serve as alternatives for heavy metal removal. It has been shown to be more efficient, less expensive, and eco-friendly than conventional clean-up methods [53,54]. Heavy-metal tolerant bacteria have been successfully used for bioremediation. For instance, metal-resistant bacteria have been shown to promote hyperaccumulating plant growth by nitrogen fixation, mineral solubilization, phytohormones, and siderophores production [55] as well as antibiotic removing ability from treated sewage effluents has been also discussed [53].

In this work, we isolated 138 bacterial strains from Amazonian cacao rhizosphere naturally cultivated in fields with high levels of cadmium soil content. We found a significant group of strains capable of surviving up to 300 ppm (88.41%). Even at 350 ppm, 36.96% of the population showed visible growth suggesting a natural ability to tolerate cadmium-contaminated environments. The morphological characterization showed that more than 70% was made up of gram-positive bacteria with a dominance of circular shape, convex elevation, absence of pigmentation, and colonies with entire margins. Similar results were shown by [28], who reported a 62% dominance of gram-positive bacteria in a cadmium-tolerant bacteria related to *T. cacao*. Conversely, [56,57] reported a strong dominance of gram-negative bacteria (over 88%). This suggests that the microbial population diversity associated with Cd-alleviation is highly variable. The microbial communities are strongly influenced by environmental factors such as pH, temperature, or nutrient availability which in agriculture is closely related to agricultural management practices [13,58,59]. Similarly, heavy-metal contamination can cause serious changes in the composition of bacterial communities and their activity in the soil [60,61]. Although this work has not analyzed the bacterial community composition, some studies in heavy-metal contaminated soils have shown a decrease in microbial diversity, indicating a detrimental effect on the establishment of the bacterial community structure and functional composition [62,63]. Therefore, the predominance of some specific bacterial groups in heavy metal-contaminated soils would not be strange. However, further characterization is required to determine the cadmium-tolerant microbial community in these soils.

According to the tolerance assay, four bacterial strains, including low-tolerant strain T3C1, were selected for further analysis. The strains S1C2, R1C2, and T3C1 were identified as *Bacillaceae* family members. They are grouped with *Bacillus weihenstephanensis* DMS11821^T^, B. *pumillus* NRRL NRS 272^T^, and *Peribacillus simplex* DSM1321^T^, respectively. Several *Bacillus* strains related to plants have been shown to be tolerant to heavy metals. Moreover, its application to cadmium-contaminated soils seems to alleviate the harmful effects and improve plant growth [64,65]. Some *Bacillus* strains have been isolated from *T. cacao*. Refs. [66,67,68] and may be predominant at high cadmium concentrations [56]. This is probably because they produce spores that offer them the ability to tolerate a wide range of biotic and abiotic stress [69]. *Peribacillus* (formerly *Bacillus*) is a novel genus recently proposed within the *Bacillaceae* family [51] that harbors species, such as *P. simplex* (previously known as Bacillus simplex), able to remove significant quantities of cadmium from solution [70]. Our strain, V3C3, was related to the *Pseudomonas putida* group. Some *Pseudomonas* strains have been found in association with cacao cultivated under cadmium-contaminated soils [28,57]. However, little is known about their effect on cadmium stress alleviation for cacao trees.

Some confusion may arise from the terms “biosorption” and “bioaccumulation”. Biosorption mechanisms are related to the uptake by dead biomass, while bioaccumulation is defined as the uptake of toxicants by living cells [71]. Here, we analyzed the bioaccumulation capacity of selected cadmium-hypertolerant bacteria. For that, the hypertolerant strains R1C2, S1C2, and V3C3 were tested under cadmium toxicity conditions. They were able to grow over 10^8^ CFU/mL in all conditions. Also, their cadmium removal efficiency and their bioaccumulation capacity were measured. Although all the strains showed removal efficiency of over 39%, the strains *Bacillus* sp. R1C2 and *Pseudomonas* sp. V3C3 showed significant removal efficiency at 200 and 300 ppm of CdCl_2_ compared with S1C2. Various *Bacillus* and *Pseudomonas* strains have been shown to metabolize a wide range of heavy metal contaminants. For example, *Bacillus* spp., gram-positive spore-forming bacteria, thrives under severe environmental stress. Lately, it has been investigated for its role in the bioremediation of metal-contaminated environments by biosorption, and bioaccumulation, among many other mechanisms [72]. Studies carried out by [73] have shown that *B. megaterium*, was able to remove the 24.09% of cadmium in a liquid solution of CdCl_2_ at 1951 mg/kg. However, this capacity gets reduced in combination with other microorganisms such as *Rhizopus stolonifer*. Conversely, a synergistic effect can be seen with the addition of amendments. For example, the Cd^2+^ removal rates in three composite systems (biochar and *B. subtilis*) were higher than those in single systems [74]. Removal efficiency capacity was also reported for *B. cereus* RC-1, which showed at least 39.14% of cadmium reduction in aqueous solution [75]. Similarly, exceptional removal efficiency was reported for *Pseudomonas aeruginosa* which can remove 99% of cadmium under aerobic conditions [76]. Similar findings were done for *Pseudomonas* sp. K32, which showed a Cd^2+^ bioaccumulation potential of at least 90%. Moreover, it was shown to improve Indol-acetic acid production, nitrogen fixation, and phosphate solubilization properties under Cd^2+^ stress [77].

Additionally, all the strains showed cell cadmium bioaccumulation capacity among all the treatments. However, *Pseudomonas* sp. V3C3 did not seem to remove cadmium by bioaccumulation, suggesting the use of different detoxifying mechanisms by the strain. Although V3C3 showed better removal efficiency, non-significant differences in the efficiency of Cd^2+^ intracellular bioaccumulation were found with increasing concentrations of Cd^2+^. Similar behavior has been reported for *Pseudomonas* sp. IV111-14 [78]. This might suggest an alternative mechanism independent of passive sorption employed by V3C3 to reduce the cadmium content in the medium. Proteins and polysaccharides in extracellular polymeric substances (EPS) are the main reasons for resistance to external heavy metals; through ion exchange, complexation, and surface precipitation, they can be produced to decrease heavy metal ions concentration in aqueous solutions [79]. For example, the high Cd-resistant strain *Pseudomonas* sp. K32 showed Cd-induced exopolysaccharide secretion for metal chelation onto cells [77]. Likewise, under the optimal cadmium stress conditions, *Pseudomonas aeruginosa* EPS, especially the proteins, had the most significant increase, which was more than 40% compared with the non-stress treatment [80].

Although biosorption seems to be a major removal mechanism for Bacillus [72,75,81], we found an excellent cadmium intracellular accumulation potential for R1C2 and S1C2 strains. In particular, the strain Bacillus sp. R1C2 increased, by almost four-fold, the cadmium accumulation with an increase in Cd^2+^ initial concentration. The intracellular accumulation of Bacillus cereus RC-1 was the predominant mechanism for cadmium removal at lower metal concentrations (below 20 mg/L). However, unlike our results, it can be overshadowed by extracellular adsorption at higher concentrations [82,83] Furthermore, it might be accompanied by Na^+^ uptake, suggesting the Cd-efflux system depends on Na^+^ uptake [84].

The removal efficiency of Cd^2+^ may vary significantly among strains, most likely due to differing Cd^2+^ resistance mechanisms [32,78]. While some strains avoid or reduce the entrance of heavy metals through extracellular entrapment, formation of complexes, or redox reactions, others accumulate it intracellularly [33]. Regardless of the mechanism employed, the expression of several genes located on plasmids or bacterial chromosomes is involved. The intracellular accumulation involves the Cd^2+^ transport into the cytoplasm [84]. In this regard, Cd^2+^ is sequestered by metal-binding proteins such as bacterial metallothioneins (bacMTs). These low-molecular cys-rich cytosolic polypeptides are able to bind to a range of heavy metals [85]. At this time, since their initial characterization in *Pseudomonas putida*, only three bacMTs have been investigated to date, including the recently described PsdMTs in the *Pseudomonas* group [86,87,88].

In the bacterial inoculation assay, we only found significant differences in growth for the strain V3C3, which significantly reduced the stem fresh weight (FW_stem_). We also observed a positive trend in all variables (FW_stem_, FW_roots_, DW_stem_, and DW_root_) with the increment of Cd^2+^ concentration for some strains. Bacterial inoculation has been proven to promote cacao growth. *Enterobacter cloacae* and *Bacillus subtilis* endophytes isolated from healthy adult cacao plants were shown to improve seedling growth [67]. Similarly, several stress-tolerant *Bacillus* sp. strains showed a significant improvement in the fresh and dry weight of shoots and roots of cacao [69]. These suggest the positive effect of bacterial inoculation on cacao growth.

Surprisingly we found high detectable levels of Cd^2+^ even in the NB treatment, suggesting an extra source of cadmium in the pot. Some studies have shown that cocoa bean samples from Peruvian north regions such as Piura, Tumbes, and Amazonas exceed the allowed Cd^2+^ concentration (0.8 μg/g) [23]. Furthermore, mean values of 0.49 and 0.99 μg/g have been reported in both the testa and cotyledon of Amazonian samples, respectively [25]. In that sense, we hypothesize that the Cd^2+^ observed in the seedlings may come from the seeds themselves. Moreover, we highlight the importance of developing and using cadmium removal strategies during Amazonian cacao cultivation even in non-contaminated soils.

Despite the in vitro performance, we found variable outcomes on cacao cadmium accumulation. The *Bacillus* strains S1C2 and R1C2 showed a negative trend in stem cadmium accumulation. Even though non-significant differences were found, likely due to data dispersion, they showed a considerable Cd^2+^ reduction level. Also, we found a significant increase in Cd^2+^ accumulation by the cadmium low-tolerant strain T3C1, highlighting not only the importance of a bacterial selection for specific environmental conditions but also the complexity of plant–bacteria interactions. This complexity has been recently supported by [43], who reported a variable nodulation performance of *Lotus tenuis* co-inoculated with different consortia of rhizobia and *Pantoea eucalypti* strains. In that sense, from a biotechnological approach, plant–bacteria interaction analyses are fundamental for the correct selection of an efficient PGPR strain.

To reduce cadmium accumulation in cacao, some strategies have been evaluated. For example, different combinations and doses of soil amendments have been shown to reduce Cd^2+^ accumulation in plant tissues [89,90,91]. Likewise, a reduction in the Cd^2+^ accumulation in the stem was also seen by native arbuscular mycorrhizal fungi [92]. Similar to our findings, several studies have shown the alleviation potential of *Bacillus* strain for plants. The inoculation of different *Bacillus* strains promotes the tolerance to Cd^2+^ toxicity, germination, plant height, leaf length, number of leaves, and fresh weight of Zea mays [93]. Also, it may reduce Cd^2+^ accumulation in grains and increase the activity of several antioxidant enzymes in *Oryza sativa* L. [94,95]. Even more, some studies have attributed the Cd-alleviation to metabolic regulation by bacteria. A recent study has shown that volatile organic compounds emitted by *Bacillus* SAY09 decreased the levels of reactive oxygen species, malondialdehyde, and electrolyte leakage in *Arabidopsis* plants and improved Fe^+^ uptake and auxin production [96]. Similar antioxidant responses were reported for *Bacillus cereus* ALT1 associated with soybeans [97].

## 5. Conclusions

The Amazonian cacao rhizosphere harbors a bacterial population naturally adapted to cadmium-contaminated soils. These bacteria are mainly composed of gram-positive strains, capable of growing under a wide range of in vitro cadmium concentrations. Some of them, even at the highly toxic levels of 400 ppm of Cadmium.

Even though there is a dominance of gram-positive bacteria among the hypertolerant strains, it is possible to find species of both the *Bacillaceae* family and the *Pseudomonas* genera. These strains grow normally and may reduce the cadmium content under in vitro conditions. However, the detoxification mechanisms employed are variable. Whereas R1C2 and S1C2 *Bacillus* strains accumulate cadmium into cells, the strain *Pseudomonas* sp. V3C3 seems to employ a different cadmium removal mechanism, probably at the extracellular level.

Moreover, despite their in vitro performance, they exert highly variable outcomes on cacao cadmium accumulation. This confirms the complexity of plant–bacteria interactions previously reported. It also demonstrates the importance of carrying out evaluations in plant–bacteria interaction systems, since they are fundamental for the correct selection of bacteria with biotechnological potential as heavy metal alleviators in cacao.

Finally, even in non-contaminated soils, it is possible to detect high levels of Cd^2+^ in cocoa plants. This highlights the importance of developing remedial strategies according to the environmentally friendly needs of a crop such as Amazonian cacao.

## Figures and Tables

**Figure 1 microorganisms-10-02108-f001:**
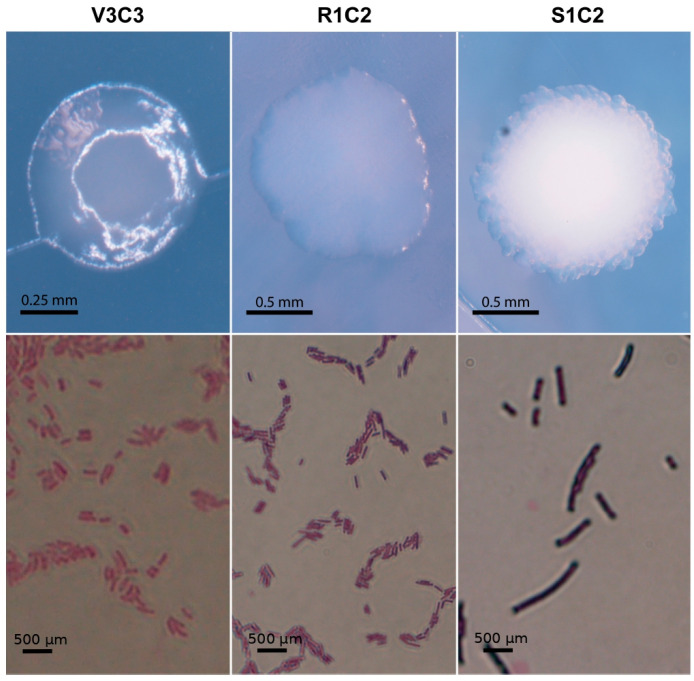
Morphological characterization of hypertolerant bacteria (HCdB). Above: Bacterial colonies photographs got by Nikon SMZ18 stereoscope coupled to DS-Ri2 camera. Below: Bacterial Gram stain microscopy got by Olympus DP74 microscope at 100×. Pictures in the same column belong to the strain indicated on the top.

**Figure 2 microorganisms-10-02108-f002:**
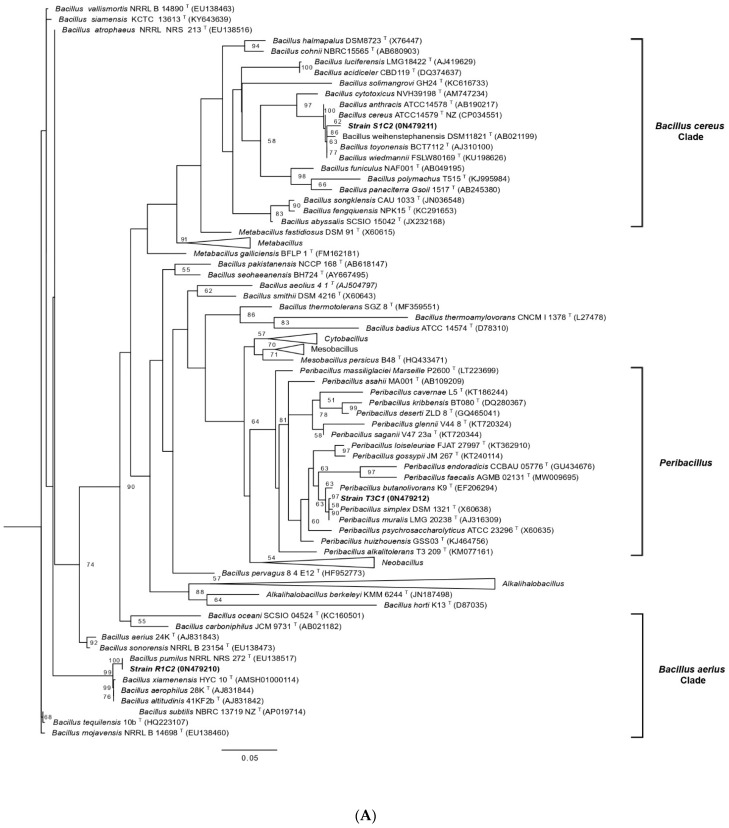
Maximum likelihood phylogenetic tree of the 16S rRNA gene for hypertolerant bacteria (HCdB) identification. The trees were constructed on the base of 16S rRNA sequences (1197 and 1395 nucleotides, respectively) of some species of *Bacillaceae* family (**A**) and *Pseudomonas* genus (**B**) reference sequences, using Kimura-2-parameter and gamma distribution models (G+I). Bootstrap values calculated for 1000 replications are indicated at the nodes. GenBank accession numbers are indicated in parentheses and bacterial clades are indicated in brackets. The complete tree of *Bacillaceae* members used is shown in Appendix A.

**Figure 3 microorganisms-10-02108-f003:**
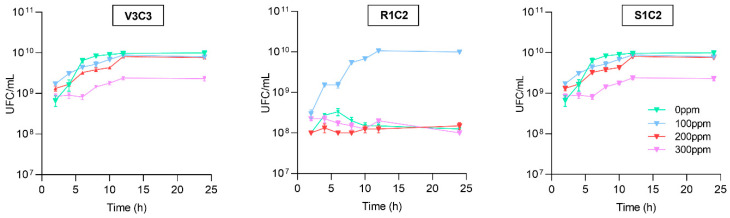
Bacterial growth kinetic characterization under cadmium conditions of selected hypertolerant bacteria. The HCdB strains were cultured in LB medium at 0, 100, 200, and 300 ppm of CdCl_2_, and the CFU/mL were quantified at 2, 4, 6, 8, 10, 12, and 24 h (h). Each point represents the means of three replicates ± SE. The graphics were performed using GraphPad software v.8.0.1 by Dotmatics, San Diego, CA, USA.

**Figure 4 microorganisms-10-02108-f004:**
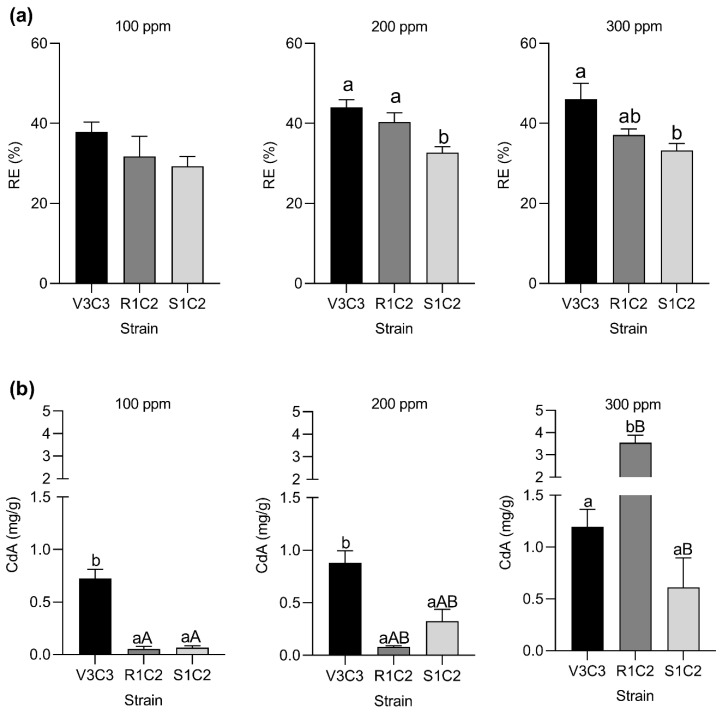
Removal efficiency and intracellular accumulation of cadmium by hypertolerant bacteria. The HCdB strains were cultured in LB medium at 0, 100, 200, and 300 ppm of CdCl_2_. Both, removal efficiency (**a**); and intracellular cadmium accumulation (**b**) were measured by Cd^2+^ quantification. The bars represent the means of 3 replicates ± SE. Data were analyzed by ANOVA followed by Tukey’s multiple-comparison test. Different letters indicate statistically significant differences (*p* ≤ 0.05). Lowercase letters represent comparisons between strains for each culture condition, while capital letters represent statistical comparisons for CdA among increasing concentrations for each strain in both RE (%) and CdA. The graphics were performed using GraphPad software v.8.0.1 by Dotmatics, San Diego, CA, USA.

**Figure 5 microorganisms-10-02108-f005:**
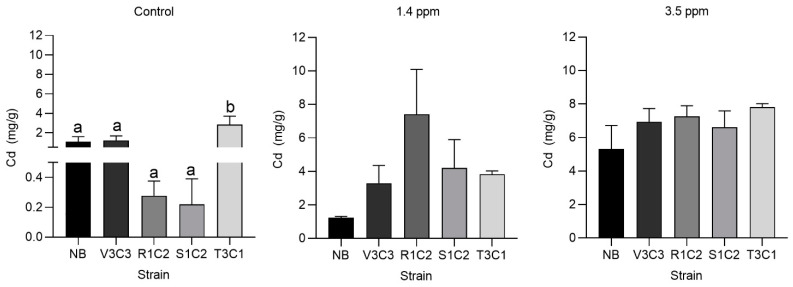
Effect of selected hypertolerant bacteria in stem cadmium accumulation by *Theobroma cacao*. Bars are means of three to five biological replicates ± SE. The values were analyzed by ANOVA followed by Tukey’s multiple-comparison test. Different letters indicate statistically significant differences (*p* ≤ 0.05). The graphics were performed using GraphPad software v.8.0.1. by Dotmatics, San Diego, CA, USA.

**Table 1 microorganisms-10-02108-t001:** Phenotypic characterization of cadmium-resistant bacteria associated with cacao.

Isolate	Sample Site	Phenotype				Gram (+)	Gram (−)
		Shape	Border	Colour	Elevation		
O2C1	Aramango-O	C	E	nP	c	●	
F1C3	Aramango-F	C	E	nP	Fl	●	
H1C2	La peca-H	C	E	p	R		●
H2C1	La peca-H	C	E	nP	c	●	
H3C1	La peca-H	C	E	nP	c	●	
H4C1	La peca-H	C	E	nP	c	●	
H4C2	La peca-H	P	E	nP	c	●	
H5C1	La peca-H	C	E	nP	c		●
H5C2	La peca-H	C	E	nP	c	●	
J1C2	La peca-J	C	E	p	R	●	
J2C1	La peca-J	P	E	nP	R		●
J2C2	La peca-J	C	E	p	c		●
J3C1	La peca-J	C	E	p	c	●	
J4C1	La peca-J	I	C	nP	Fl		●
J4C2	La peca-J	C	E	nP	R	●	
J5C1	La peca-J	C	E	nP	c	●	
T2C1	La peca-T	C	E	nP	Fl		●
T2C2	La peca-T	I	C	nP	c	●	
T3C1	La peca-T	P	E	nP	R	●	
T3C2	La peca-T	C	E	p	c	●	
T4C1	La peca-T	C	E	p	c	●	
T4C2	La peca-T	I	C	nP	Fl	●	
T5C1	La peca-T	C	E	nP	R	●	
T5C2	La peca-T	C	E	nP	c	●	
V3C3	Copallin-V	C	E	nP	Fl		●
S1C2	Copallin-S	I	C	nP	R	●	
S5C1	Copallin-S	C	E	p	c		●
Z5C1	Copallin-Z	C	E	nP	c		●
C1C1	Copallin-C	C	E	p	c		●
C1C2	Copallin-C	C	E	p	c		●
C2C1	Copallin-C	P	C	nP	R	●	
C3C1	Copallin-C	C	E	nP	R		●
C5C1	Copallin-C	C	E	nP	c	●	
M1C1	Copallin-M	I	C	nP	Fl	●	
M1C2	Copallin-M	C	E	p	R	●	
M2C1	Copallin-M	P	E	nP	R	●	
M3C1	Copallin-M	P	E	nP	R	●	
M4C1	Copallin-M	C	E	p	c	●	
M4C2	Copallin-M	I	L	nP	R	●	
M5C2	Copallin-M	C	E	p	c	●	
E1C2	Copallin-E	C	E	p	c	●	
E2C2	Copallin-E	C	E	p	c	●	
E3C1	Copallin-E	C	E	p	R	●	
R1C1	Copallin-R	C	E	nP	R	●	
R1C2	Copallin-R	I	L	nP	c	●	
R2C1	Copallin-R	C	E	nP	c	●	
R2C2	Copallin-R	C	E	p	c	●	
R3C1	Copallin-R	C	E	nP	R		●
R3C2	Copallin-R	P	E	p	R	●	
R4C1	Copallin-R	C	E	nP	c	●	
R5C2	Copallin-R	F	F	nP	Fl		●
R5C3	Copallin-R	P	E	p	R	●	

The colony in vitro phenotypic variables are C: circular, I: Irregular, P: Punctiform, F: Filamentous, E: Entire, C: Curly, L: Lobed, p: Pigmented, nP: No pigmented, c: Convex, R: Raised, Fl: Flat.

**Table 2 microorganisms-10-02108-t002:** Plant growth variables of *Theobroma cacao* inoculated with *Pseudomonas* and *Bacillus* strains under soil supplemented with cadmium.

Variable	Strain				
	NB	V3C3	R1C2	S1C2	T3C1
	Control				
**FW_stem_**	5.7 ± 1.61	5.64 ± 1.61	7.13 ± 1.8	7.3 ± 2.08	7.97 ± 2.08
**Fw_root_**	1.96 ± 0.51	2.52 ± 0.51	2 ± 0.57	2.57 ± 0.66	1.77 ± 0.66
**DW_stem_**	1.96 ± 0.47	2.07 ± 0.47	2.18 ± 0.52	2.2 ± 0.6	2.51 ± 0.6
**Dw_root_**	0.52 ± 0.16	0.65 ± 0.16	0.49 ± 0.18	0.61 ± 0.21	0.74 ± 0.21
	1.4 ppm				
**FW_stem_**	6.8 ± 2.83	10.88 ± 2.0	5.77 ± 2.31	13.12 ± 1.79	9.6 ± 2
**Fw_root_**	2.05 ± 0.65	2.78 ± 0.46	1.7 ± 0.53	2.52 ± 0.41	2.23 ± 0.46
**DW_stem_**	2.15 ± 0.89	3.2 ± 0.63	1.71 ± 0.72	4.1 ± 0.56	3.23 ± 0.63
**Dw_root_**	0.25 ± 0.17	0.78 ± 0.12	0.42 ± 0.14	0.7 ± 0.11	0.65 ± 0.12
	3.5 ppm				
**FW_stem_**	13.55 ± 1.54 ^b^	4.87 ± 1.54 ^a^	9.58 ± 1.34 ^ab^	13.55 ± 1.34 ^b^	11.9 ± 1.89 ^ab^
**Fw_root_**	2.68 ± 0.54	1.95 ± 0.54	1.83 ± 0.54	2.72 ± 0.48	2.07 ± 0.62
**DW_stem_**	3.52 ± 0.67	1.75 ± 0.78	2.9 ± 0.67	3.83 ± 0.6	2.67 ± 0.78
**Dw_root_**	0.86 ± 0.17	0.53 ± 0.17	0.52 ± 0.17	0.76 ± 0.15	0.61 ± 0.19

The variables are Fw_stem_: Stem fresh weight, Fw_root_: Root fresh weight, Dw_stem_: Stem dry weight, and Dw_root_: Root dry weight. Values are means of 3–5 biological replicates ± SE. The values were analyzed by ANOVA followed by Tukey´s multiple-comparison test. Different letters indicate statistically significant differences (*p* ≤ 0.05).

## Data Availability

Data sharing is not applicable.

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
