# Peer review of "Bioremediation Potential of Native Bacillus sp. Strains as a Sustainable Strategy for Cadmium Accumulation of Theobroma cacao in Amazonas Region"

_microorganisms, 2022, doi:10.3390/microorganisms10112108_

Round 1
Reviewer 1 Report
This is a relevant study on a sensitive issue for many cocoa producers who need solutions to the cadmium problem. I appreciate the effort put forth by the authors in their study however I think they should tone down their claims of a possible reduction of cadmium in the plants due to their strains (fourth paragraph in page 15) as that is based only on what they observe in a control treatment for which they are not sure where the source of cadmium is coming from. If one looks at what happens in the cadmium treatments (figure 5), one can say that there is a tendency for the strains to increase the cadmium content in the plant. Moreover, the title of the article leads one to think that a solution to the problem has been found when this is not the case, so the title should also be modified, for example to something like "Exploring the potential of native Bacillus sp. strains as a ....".
Author Response
RESPONSE TO REVIEWER 1 COMMENTS
POINT 1: This is a relevant study on a sensitive issue for many cocoa producers who need solutions to the cadmium problem. I appreciate the effort put forth by the authors in their study however I think they should tone down their claims of a possible reduction of cadmium in the plants due to their strains (fourth paragraph on page 15) as that is based only on what they observe in a control treatment for which they are not sure where the source of cadmium is coming from. If one looks at what happens in the cadmium treatments (figure 5), one can say that there is a tendency for the strains to increase the cadmium content in the plant. Moreover, the title of the article leads one to think that a solution to the problem has been found when this is not the case, so the title should also be modified, for example to something like "Exploring the potential of native Bacillus sp. strains as a ....".
RESPONSE:
We have taken into account the thoughtful suggestion of the reviewer. In that sense, we have modified the title in order to be accurate with the findings. The previous title: "Native Bacillus strains as a sustainable strategy for cadmium bioremediation to Theobroma cacao in Amazonas region" has been changed to “Bioremediation potential of native Bacillus sp. strains as a sustainable strategy for cadmium accumulation of Theobroma cacao in Amazonas region".
Please see the attachment

Reviewer 2 Report
The article presents an intresting results on potential application of seleceted bacterial strains to decrease negative effect of Cd contained in soil on cacao plants.
The article should be improved:
Page 1. “138 rhizosphere isolations” – isolates?
Page 1. “Moreover, despite their (?) in vitro features, they exerted highly variable outcomes on stem cadmium accumulation.” The sentence is not clear.
Page 1 “In particular, we found a negative trend for S1C2 and R1C2.” What is the negative trend?
Page 3, 4, 10 and other: UFC/mL --> colony forming units (CFU/mL)?
Page 3 “sodium chloride solution (NaCl2) at 0.85%” – 0.85% (w/w) NaCl solution?
Page 3 “Luria-Bertani (LB) medium” – liquid/agar medium (also, include agar%)?
Page 3 “According to similarity, representative type strains sequences belonging to Pseudomonas genus and Bacillaceae family were included.” The sentence is not clear.
Page 4 “with NaCl2 at 0.85%” --> 0.85% (w/w) NaCl solution?
Page 4 “at 3 ppm (micro toxicity)” --> 3 ppm – Is it Cd content in the soil or in the medium?
Page 4-5. “The latter group was classified as hypertolerant-cadmium bacteria (HCdB) and were selected for the removal efficiency and the cadmium accumulation capacity tests.” Could you explain the selection of hypertolerant-cadmium bacteria (HCdB) as the most promising for Cd removal. High Cd resistance may be based not only on the ability of Cd accumulation, but also on other mechanisms.
Figure 3. – Use color graphs as the curves are difficult to distinguish from each other.
Figures 3, 4, 5 – Remove designations Figure 03, Figure 04, Figure 05 from the top of the figures.
Figure 4. Clarify designations “a”, “b”, “ab”, “Aa”, “aAB”, etc. on the diagrams.
Table 2. Clarify designations “a”, “b”, “ab” in the footnote.
Page 15. “In conclusion……” – Make separate section “CONCLUSIONS”
Table S1. - Specify the meaning of the numbers in the columns. Is this the number of isolates capable of growing at different concentrations of Cd in the medium?
In general, opposite effects of the strains studied on Cd accumulation at different Cd contents in soil should be explained and noted in the Abstract.
Author Response
RESPONSE TO REVIEWER 2 COMMENTS
Point 1: “138 rhizosphere isolations” – isolates?
Response: Done. 138 rhizosphere isolations were replaced by 138 rhizosphere isolates.
Point 2: “Moreover, despite their (?) in vitro features, they exerted highly variable outcomes on stem cadmium accumulation.” The sentence is not clear.
Response: Clarification was added. The sentence refers to cadmium removal behavior under in vitro conditions. To clarify, it was replaced by “Moreover, despite their cadmium reduction performance under in vitro conditions, they exerted highly variable outcomes on stem cadmium accumulation”.
Point 3: “In particular, we found a negative trend for S1C2 and R1C2.” What is the negative trend?
Response: It refers to the plant cadmium accumulation reduction by the strains. The sentence was modified. The sentence “In particular, we found a negative trend for S1C2 and R1C2” was modified by “While S1C2 and R1C2 showed a considerable reduction of Cd content in cacao stems, V3C3 did not exert any effect on Cd content”.
Point 4: Page 3, 4, 10 and other: UFC/mL --> colony forming units (CFU/mL)?
Response: Done
Point 5: “sodium chloride solution (NaCl2) at 0.85%” – 0.85% (w/w) NaCl solution?
Response: The sentence was corrected
Point 6: “Luria-Bertani (LB) medium” – liquid/agar medium (also, include agar%)?
Response: Done. Luria-Bertani (LB) medium was replaced by Luria-Bertani (LB) agar medium
Point 7: “According to similarity, representative type strains sequences belonging to Pseudomonas genus and Bacillaceae family were included.” The sentence is not clear.
Response: This sentence is referred to the preliminary search performed with the query in order to select the reference sequences for the phylogeny analysis. To clarify, the sentence was modified to “According to the preliminary similarity test, representative type strains sequences belonging to Pseudomonas genus and Bacillaceae family, were included”.
Point 8: “with NaCl2 at 0.85%” --> 0.85% (w/w) NaCl solution?
Response: The sentence was corrected
Point 9 “at 3 ppm (micro toxicity)” --> 3 ppm – Is it Cd content in the soil or in the medium?
Response: It refers to the medium. To clarify, the sentence was modified to "in LB agar medium supplemented with 3 ppm of CdCl2 (micro toxicity)". Moreover, “micro toxicity condition” was added to section 2.2 of Material and Methods.
Point 10: “The latter group was classified as hypertolerant-cadmium bacteria (HCdB) and were selected for the removal efficiency and the cadmium accumulation capacity tests.” Could you explain the selection of hypertolerant-cadmium bacteria (HCdB) as the most promising for Cd removal. High Cd resistance may be based not only on the ability of Cd accumulation but also on other mechanisms.
Response: The HCdB was selected because, as we have already mentioned in the manuscript, the Cadmium tolerant bacteria (CdtB) have the capacity to interact with the Cadmium present in the soil. Not only through resistance mechanisms, but they can also metabolize and use it as a source of energy to activate various mechanisms such as bioaccumulation or biosorption. Therefore, tolerance was our first selection criteria to select a promising group of bacteria.
On the other hand, indeed the tolerance may be done by several mechanisms. As you can see in the results section, our strains use different cadmium removal strategies. In particular, despite that the strain Pseudomonas V3C3 was the most efficient under in vitro conditions, it seems to use different mechanisms than bioaccumulation. However, when it was tested with CCN51 it didn't show any effect. In that sense, we highlight the importance of plant-bacteria interaction tests for promising bacteria selection.
Moreover, it should be said that the focus of this job is looking for a better plant cadmium removal strain, indistinctly from its tolerance mechanism. The in vitro tests were complementary for a better understanding of the response strategies employed by the strains.
Point 11. Use color graphs as the curves are difficult to distinguish from each other.
Response: Done
Point 12. Figures 3, 4, 5 – Remove designations Figure 03, Figure 04, Figure 05 from the top of the figures.
Response: Done
Point 13: Figure 4. Clarify designations “a”, “b”, “ab”, “Aa”, “aAB”, etc. on the diagrams.
Response: Done. The paragraph, “Different letters indicate statistically significant differences (p ≤ 0.05). Lowercase letters represent comparisons between strains for each culture condition, while capital letters represent statistical comparisons for CdA among increasing concentrations for each strain in both RE (%) and CdA”, was added.
Point 14 Table 2. Clarify designations “a”, “b”, “ab” in the footnote.
Response: Done. The paragraph “Values are means of 3-5 biological replicates ± SE. The values were analyzed by ANOVA followed by Tukey´s multiple-comparison test. Different letters indicate statistically significant differences (p ≤ 0.05)”, was added.
Point 15 : Page 15. “In conclusion……” – Make separate section
“CONCLUSIONS”
Response: Done.
Point 16: Table S1. - Specify the meaning of the numbers in the columns. Is this the number of isolates capable of growing at different concentrations of Cd in the medium?
Response: Yes, the values refer to the numbers of tolerant isolations at different Cd conditions. The specification was added.
Point 17: In general, opposite effects of the strains studied on Cd accumulation at different Cd contents in soil should be explained and noted in the Abstract.
Response: Done.
Please see the attachment.

Reviewer 3 Report
Dear authors,
Manuscript microorganisms-1922463 entitled "Natives Bacillus sp. strains as a sustainable strategy for cadmium bioremediation to Theobroma cacao in Amazonas region" and authored by "Marielita Arce-Inga , Alex Ricardo González-Pérez , Elgar Hernandez-Diaz , Beimer Chuquibala-Checan , Antony Chavez-Jalk , Kelvin James Llanos-Gomez , Santos Triunfo Leiva-Espinoza , Segundo Manuel Oliva-Cruz and Liz Marjory Cumpa-Velasquez" targets a hot topic that fits well with the scope of the journal and that is highly relevant for the journal readers. While the design of the study is accurate and the experiments nicely conducted few points needs the authors attention. Therefore I could not recommand the manuscript for publication before these points addressed:
1. The title : the title does not encourage readers to really read the paper I suggest changing the title and stressing the findings of the manuscript in the new title. This is very important for the potential of citations of this manuscript.
2. Introduction section : there is no serious discussion about cadmium tolerant PGPR. This is the main point of your manuscript so please dedicate a section to describe state of the art research in the field.
3. Introduction section : there is no serious discussion about cadmiun tolerance mechanism in PGPR cadmium tolerant. This is a critical issue to move your study from a descriptive study to a study that targets mechanistic aspects for alleviation of cadmium toxicity and cadmium bioremediation for theobroma cacao.
4. Material and methods section: please be precise and describe what you call rhizosphere soil. Is it the soil that sticks to plant roots?
5. Material and methods section: have you checked any features of the PGPR bacteria? do they produce auxin? siderophores? do they solubilize phosphate?
6. Material and methods section: have you checked any potential for plant growth promotion of your PGPR bacteria?
7. Results section: please improve the resolution of your trees they are actually not readable and therefore meaningless.
8. in discussion section you claim "In that sense, we hypothesize that the Cd2+ observed in the seedlings may come from the seeds themself. Moreover, we highlight the importance of developing and using cadmium removal strategies during Amazonian cacao cultivation even under non-contaminated soils." have you measured cadmiun concentrations in of the seedlings?
I am really looking towards reading an improved version of this manuscript that addresses all comments and that I could recommend for publication
Best regards
Author Response
RESPONSE TO REVIEWER 3 COMMENTS
Point 1: The title: the title does not encourage readers to really read the paper I suggest changing the title and stressing the findings of the manuscript in the new title. This is very important for the potential of citations of this manuscript.
Response: We have taken into account the thoughtful suggestion of the reviewer. In that sense, we have modified the title in order to be accurate with the findings. The title "Native Bacillus strains as a sustainable strategy for cadmium bioremediation to Theobroma cacao in Amazonas region" has been changed to “Bioremediation potential of native Bacillus sp. strains as a sustainable strategy for cadmium accumulation of Theobroma cacao in Amazonas region".
Point 2: Introduction section: there is no serious discussion about cadmium tolerant PGPR. This is the main point of your manuscript so please dedicate a section to describing state of the art research in the field.
Response: We appreciate the reviewer’s suggestion. We have dedicated a section to discuss Cadmium tolerant bacteria (CdTB).
Point 3: Introduction section: there is no serious discussion about the cadmium tolerance mechanism in PGPR cadmium tolerant. This is a critical issue to move your study from a descriptive study to a study that targets mechanistic aspects for alleviation of cadmium toxicity and cadmium bioremediation for theobroma cacao.
Response: We appreciate the reviewer’s suggestion. We have dedicated a section to Cadmium tolerant bacteria (CdTB) mechanisms overview. Moreover, tolerance mechanisms can be found in the Discussion section, also.
Point 4: Material and methods section: please be precise and describe what you call rhizosphere soil. Is it the soil that sticks to plant roots?
Response: We call rhizosphere to the soil layer adjacent to the root. Definition was added
Point 5: Material and methods section: have you checked any features of the PGPR bacteria? do they produce auxin? siderophores? do they solubilize phosphate?
Response: Yes, we have tested Zinc and phosphorus solubilization, as well as siderophore production. However, we didn't include the phenotypic tests in the results section, because we didn’t see plant growth promotion effects on the plant growing variables. Moreover, we didn't measure macro and micronutrients except for Cadmium content. So we were not able to discuss nutritional status effects. Nevertheless, we have already added the PGPR tests in section 2.8.
Point 6: Material and methods section: have you checked any potential for plant growth promotion of your PGPR bacteria?
Response: Yes. We have performed a plant-bacteria interaction test described in sections 2.9 and 3.4. The results of the plant growing variables are shown in Table 2. According to fresh (FW) and dry weight (DW) no plant growth promotion effect was observed, taking into account that there is a direct relationship between the dry matter content and nutritional status
Point 7: Results section: please improve the resolution of your trees they are actually not readable and therefore meaningless.
Response: Done.
Point 8 : In discussion section you claim "In that sense, we hypothesize that the Cd2+ observed in the seedlings may come from the seeds themself. Moreover, we highlight the importance of developing and using cadmium removal strategies during Amazonian cacao cultivation even under non-contaminated soils." have you measured cadmium concentrations in of the seedlings?
Response: No, we didn't measure cadmium in the seeds or the seedlings. The cadmium content test was done just in the soil and the shoots (at the end of the experiment). The extra content of Cadmium found even in non-cadmium supplemented treatment was not expected. In that sense, we can’t precise about the source of them. However, our hypothesis was based on the cadmium present in the beans, testa, or cotyledon previously reported in the literature, especially in the recent study performed by Oliva et al 2020 in the Amazonas Peruvian region.
Round 2
Reviewer 3 Report
I can now recommend manuscript for publication